# Rethinking Counterfactual Explanations as Local and Regional Counterfactual Policies

## Abstract

Among the challenges not yet resolved for Counterfactual Explanations (CE), there are stability, synthesis of the various CE and the lack of plausibility/sparsity guarantees. From a more practical point of view, recent studies show that the prescribed counterfactual recourses are often not implemented exactly by the individuals and demonstrate that most state-of-the-art CE algorithms are very likely to fail in this noisy environment. To address these issues, we propose a probabilistic framework that gives a sparse local counterfactual rule for each observation: we provide rules that give a range of values that can change the decision with a given high probability instead of giving diverse CE. In addition, the recourses derived from these rules are robust by construction. These local rules are aggregated into a regional counterfactual rule to ensure the stability of the counterfactual explanations across observations. Our local and regional rules guarantee that the recourses are faithful to the data distribution because our rules use a consistent estimator of the probabilities of changing the decision based on a Random Forest. In addition, these probabilities give interpretable and sparse rules as we select the smallest set of variables having a given probability of changing the decision. Codes for computing our counterfactual rules are available, and we compare their relevancy with standard CE and recent similar attempts.

## 1 Introduction

In recent years, many explanations methods have been developed for explaining machine learning models, with a strong focus on local analysis, i.e., generating explanations for individual prediction (see [Molnar, 2022] for a survey). Among this plethora of methods, one of the most prominent and active techniques are Counterfactual Explanations [Wachter et al., 2017b]. Unlike popular local attribution methods, e.g., SHAP [Lundberg et al., 2020] and LIME [Ribeiro et al., 2016], which highlight the importance score of each feature, Counterfactuals Explanations (CE) describe the smallest modification to the feature values that changes the prediction to a desired target. Although CE are intuitive and user-friendly by giving recourse in some scenarios (e.g., loan application), they have many shortcomings in practice. Indeed, most counterfactual methods rely on a gradient-based algorithm or heuristics approaches [Karimi et al., 2020b], thus can fail to identify the most natural explanations and lack guarantees. Most algorithms either do not guarantee sparse counterfactuals (changes in the smallest number of features) or do not generate in-distribution samples (see [Verma et al., 2020, Chou et al., 2022] for a survey on counterfactuals methods). Although some works [Parmentier and Vidal, 2021, Poyiadzi et al., 2019, Looveren and Klaise, 2019] try to solve the plausibility/sparsity problem, the suggested solutions are not entirely satisfactory.

In another direction, many papers [Mothilal et al., 2020, Karimi et al., 2020a, Russell, 2019] encourages the generation of diverse counterfactuals in order to find actionable recourse [Ustun et al., 2019]. Actionability is a vital desideratum, as some features may be non-actionable, and generating many

counterfactuals increases the chance of getting actionable recourse. However, the diversity of CE makes the explanations less intelligible, and the synthesis of various CE or local explanations, in general, is yet to be comprehensively solved [Lakkaraju et al., 2022]. In addition, recently Pawelczyk et al. [2022] highlights a new problem of local CE called: *noisy responses to prescribed recourses*. Indeed, in real-world scenarios, some individuals may not be able to implement exactly the prescribed recourses, and they show that most CE methods fail in this noisy environment. Therefore, we propose to reverse the usual way of explaining with counterfactual by computing *Counterfactual rules*. We introduce a new line of counterfactuals: we build interpretable policies for changing a decision with a given probability that ensure the stability of the deduced recourse. These policies are optimal (in sparsity) and faithful to the data distribution. Their computation comes with statistical guarantees as they use a consistent estimator of the conditional distribution. Our proposal is to find a general policy or rule that permits changing the decision while fixing some features instead of generating many counterfactual samples. One of the main challenges is to identify the (minimal) set of features that provide the best promising directions for changing the decision to the desired output. We also show this approach can be extended for finding a collection of regional counterfactuals, such that we have a global counterfactual policy for analyzing a model. An example of the counterfactual rules that we introduce is given in figure 1.

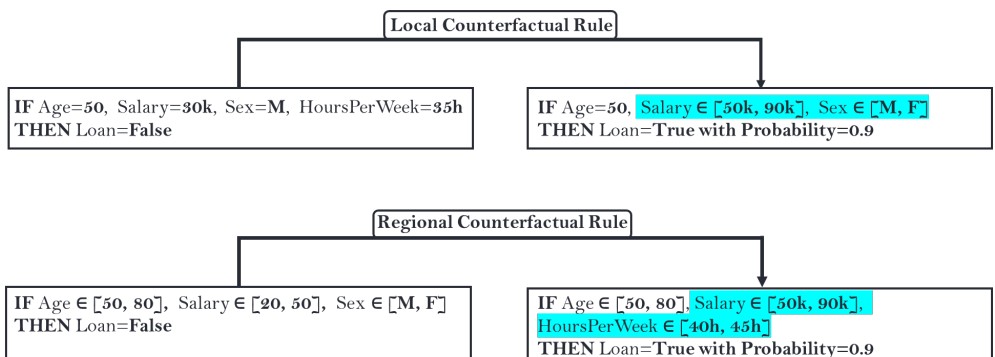

Figure 1: Illustration of the local and regional Counterfactuals Rules that we introduced on a dataset with 4 variables: Age, Salary, Sex, and HoursPerWeek. The Counterfactual Rules define intervals on the minimal subset of features to change the decision of a model prediction in the local counterfactual rule or the decision of a rule that applies on a sub-population in the regional counterfactual rule. In Blue, we have the proposed rules to change the decision.

## 2  Motivation and Related works

Most of the methods that propose Counterfactuals Explanations are based on the approach of the seminal work of Wachter et al. [2017a]: the counterfactuals are generated by optimizing a cost, but this procedure does not account directly the plausibility of the counterfactual examples (see [Verma et al., 2020] for classification of CE methods). Indeed, a major shortcoming is that the adverse decision needed for obtaining the counterfactual is not designed to be feasible or representative of the underlying data distribution. However, some recent studies proposed ad-hoc plausibility constraint in the optimization, using for instance an outlier score [Kanamori et al., 2020], an Isolation Forest [Parmentier and Vidal, 2021] or a density-weighted metrics [Poyiadzi et al., 2019] to generate in-distribution samples. In another direction, Looveren and Klaise [2019] proposes to use an autoencoder that penalizes out-of-distribution candidates. Instead of relying on ad-hoc constraints, we propose CE that gives plausible explanations by design. Indeed, for each observation, we identify the variables and associated ranges of values that have the highest probability of changing the prediction. We can compute this probability with a consistent estimator of the conditional distribution $P(Y|\boldsymbol{X}_S)$. As a consequence, the sparsity of the counterfactuals is not encouraged indirectly by adding a penalty term ($\ell_0$ or $\ell_1$) as existing works [Mothilal et al., 2020]. Our approach is inspired by the concept of *Same Decision Probability (SDP)* (introduced in [Chen et al., 2012]) that can be used for identifying the smallest subset of features to guarantee (with a given probability) the stability of a prediction. This minimal subset is called *Sufficient Explanations*. In [Amoukou and Brunel, 2021], it has been shown

that the *SDP* and the *Sufficient Explanations* can be estimated and computed efficiently for identifying important local variables in any classification and regression models. For counterfactuals, we are interested in the dual set: we want the minimal subset of features that have a high probability of changing the decision (when the other features are fixed). Another limitation of the current CE is their local nature and the multiplicity of the explanations produced. While some papers [Mothilal et al., 2020, Karimi et al., 2020a, Russell, 2019] promote the generation of diverse counterfactual samples to ensure actionable recourse, such diverse explanations should be summarized to be intelligible [Lakkaraju et al., 2022], but the compilation of local explanations is often a very difficult problem. To address this problem, we do not generate counterfactual samples, but we build a rule *Counterfactual Rules* (CR) from which we can derive counterfactuals. Contrary to classic CE which gives the nearest instances with a desired output, we find the most effective rule for each observation (or group of similar observations) that changes the prediction to the desired target. This local rule easily aggregates similar counterfactuals. For example, if $x = \{$Age=20, Salary=35k, HoursWeek=25h, Sex=M,...$\}$ with Loan=False, fixing the variables Age and Sex and changing the Salary and HoursWeek change the decision. Therefore, instead of given multiples combination of Salary and HoursWeek (e.g. 35k and 40h or 40k and 55h, ...) that result in many instances, the counterfactual rule gives the range of values: IF HoursWeek $\in$ [35h, 50h], Salary $\in$ [40k, 50k], and the **remaining features are fixed** THEN Loan=True. It can be extended at a regional scale, e.g., given a rule $\mathbf{R} = \{$IF Salary $\in$ [35k, 20k], Age $\in$ [20, 80] THEN Loan=False$\}$, the regional Counterfactual Rule (CR) could be $\{$IF Salary $\in$ [40k, 50k], HoursWeek $\in$ [35h, 50h] and the **remaining rules are fixed** THEN Loan=True$\}$. The main difference between a local and a global CR is that the Local-CR explain a single instance by fixing the remaining feature values (not used in the CR) ; while a regional-CR is defined by keeping the remaining variables in a given interval (not used in the regional-CR). Moreover, by giving ranges of values that guarantee a high probability of changing the decision, we partly answer the problem of *noisy responses to prescribed recourses* [Pawelczyk et al., 2022] so long as the perturbations are within our ranges.

Although the *Local Counterfactual Rule* is new, the *Regional Counterfactual Rule* can be related to some recent works. Indeed, Rawal and Lakkaraju [2020] proposed Actionable Recourse Summaries (AReS), a framework that constructs global counterfactual recourses in order to have a global insight of the model and detect unfair behavior. While AReS is similar to the Regional Counterfactual Rule, we emphasize some significant differences. Our methods can address regression problems and deal with continuous features. Indeed, AReS needs to discretize the continuous features, inducing a trade-off between speed and performance as noticed by [Ley et al., 2022]. Thus, too few bins result in unrealistic recourse, while too many bins result in excessive computation time. In addition, AReS uses a greedy heuristic search approach to find global recourse, which might produce sub-optimal recourse. As we have already mentioned, the changes we provide overcome these two limitations because the consistency of our counterfactual is controlled by an estimation of the probability of changing the decision, and because we favor changes of a minimum number of features. Another global CE framework has been introduced in [Kanamori et al., 2022] to ensure transparency: the Counterfactual Explanation Tree (CET) partitions the input space with a decision tree and assigns an appropriate action for changing the decision of each subspace. Therefore, it gives a unique action for changing the decision of multiple instances. In our case, we offer more flexibility in the counterfactual explanations because we provide a range of possible values that guarantee a change with a given probability. In our approach, we do not make any assumption about the cost of changing the feature nor the causal structure. If we have such information, then we can add it as additional post-processing such that it can be made more explicit and more transparent for the final user as required for trustworthy AI.

## 3 Minimal Counterfactual Rules

We assume that we have an i.i.d sample $\mathcal{D}_n = \{(\boldsymbol{X}_i, Y_i)_{i=1,...,n}\}$ such that $(\boldsymbol{X}, Y) \sim P_{(\boldsymbol{X},Y)}$ where $\boldsymbol{X} \in \mathcal{X}$ (typically $\mathcal{X} = \mathbb{R}^p$) and $Y \in \mathcal{Y}$. The output $\mathcal{Y}$ can be discrete or continuous. We want to explain the predictor $f : \mathbb{R}^p \mapsto \mathcal{Y}$, that has been learned with the dataset $\mathcal{D}_n$. We use uppercase letters for random variables and lowercase letters for their value assignments. For a given subset $S \subset [p]$, $\boldsymbol{X}_S = (X_i)_{i \in S}$ denotes a subgroup of features, and we write $\boldsymbol{x} = (\boldsymbol{x}_S, \boldsymbol{x}_{\bar{S}})$ (with some abuse of notation).

For an observation $(\boldsymbol{x}, y = f(\boldsymbol{x}))$, we have a target set $\mathscr{Y}^\star \subset \mathcal{Y}$, such that $y \notin \mathscr{Y}^\star$. For the simple case of classification problem, $\mathscr{Y}^\star = \{y^\star\}$ is the standard singleton such that $y^\star \in \mathcal{Y}$ is different of $y$. Contrary to standard approaches, our definition of the counterfactual deals also with the regression case by considering $\mathscr{Y}^\star = [a, b] \subset \mathbb{R}$; our definitions and computations of counterfactuals are the same for both classification and regression. We remind that the classic CE problem (defined only for classification) is to find a function $\boldsymbol{a} : \mathcal{X} \mapsto \mathcal{X}$, such that for all observations $\boldsymbol{x} \in \mathcal{X}$, $f(\boldsymbol{x}) \neq y^\star$, and we have $f(\boldsymbol{a}(\boldsymbol{x})) = y^\star$. With standard CE, the function is defined point-wise by solving an optimisation program. Most often $\boldsymbol{a}(\cdot)$ is not a real function, as $\boldsymbol{a}(x)$ may be in fact a collection of (random) values $\{\boldsymbol{x}_1^\star, \ldots, \boldsymbol{x}_p^\star\}$. A more recent point of view was proposed by Kanamori et al. [2022], and it defines $\boldsymbol{a}$ as a decision tree, where in each leaf $L$, the best perturbation $a_L$ is predicted and add it to all the instances $\boldsymbol{x} \in L$.

Our approach is hybrid, because we do not propose a single action for each subspace of $\mathcal{X}$ or sub-group of population, but we give sets of possible perturbations. Indeed, a *Local Counterfactual Rule* (Local-CR) for $\mathscr{Y}^\star$ and observation $\boldsymbol{x}$ (with $f(\boldsymbol{x}) \notin \mathscr{Y}^\star$) is a rectangle $C_S(\boldsymbol{x}; \mathscr{Y}^\star) = \prod_{i \in S}[a_i, b_i], a_i, b_i \in \overline{\mathbb{R}}$ such that for all perturbations of $\boldsymbol{x} = (\boldsymbol{x}_S, \boldsymbol{x}_{\bar{S}})$ obtained as $\boldsymbol{x}^\star = (\boldsymbol{z}_S, \boldsymbol{x}_{\bar{S}})$ with $\boldsymbol{z}_S \in C_S(\boldsymbol{x}; \mathscr{Y}^\star)$ and $\boldsymbol{x}^\star$ an in-distribution sample, then $f(\boldsymbol{x}^\star)$ is in $\mathscr{Y}^\star$ with a high probability.
Similarly, a *Regional Counterfactual Rule* (Regional-CR) $C_S(\boldsymbol{R}; \mathscr{Y}^\star)$ is defined for $\mathscr{Y}^\star$ and a rectangle $\boldsymbol{R} = \prod_{i=1}^d[a_i, b_i], a_i, b_i \in \overline{\mathbb{R}}$, if for all observations $\boldsymbol{x} = (\boldsymbol{x}_S, \boldsymbol{x}_{\bar{S}}) \in \boldsymbol{R}$, the perturbations obtained as $\boldsymbol{x}^\star = (\boldsymbol{z}_S, \boldsymbol{x}_{\bar{S}})$ with $\boldsymbol{z}_S \in C_S(\boldsymbol{R}, \mathscr{Y}^\star)$ and $\boldsymbol{x}^\star$ an in-distribution sample are such that $f(\boldsymbol{x}^\star)$ is in $\mathscr{Y}^\star$ with high probability.
We build such rectangles sequentially, first, we propose to find the best directions $S \subset [p]$ that offers the best probability of change. Then, we find the best intervals $[a_i, b_i], i \in S$ that change the decision to the desired target. A central tool in this approach is the Counterfactual Decision Probability.

**Definition 3.1. Counterfactual Decision Probability (CDP).** The Counterfactual Decision Probability of the subset $S \subset [\![1, p]\!]$, w.r.t $\boldsymbol{x} = (\boldsymbol{x}_S, \boldsymbol{x}_{\bar{S}})$ and the desired target $\mathscr{Y}^\star$ (s.t. $f(\boldsymbol{x}) \notin \mathscr{Y}^\star$) is

$$CDP_S(\mathscr{Y}^\star; \boldsymbol{x}) = P(f(\boldsymbol{X}) \in \mathscr{Y}^\star | \boldsymbol{X}_{\bar{S}} = \boldsymbol{x}_{\bar{S}}).$$

The $CDP$ of the subset S is the probability that the decision changes to the desired target $\mathscr{Y}^\star$ by sampling the features $\boldsymbol{X}_S$ given $\boldsymbol{X}_{\bar{S}} = \boldsymbol{x}_{\bar{S}}$. It is related to the Same Decision Probability $SDP_S(\mathscr{Y}; \boldsymbol{x}) = P(f(\boldsymbol{X}) \in \mathscr{Y} | \boldsymbol{X}_S = \boldsymbol{x}_S)$ used in [Amoukou and Brunel, 2021] for solving the dual problem of selecting the most local important variables for obtaining and maintaining the decision $f(\boldsymbol{x}) \in \mathscr{Y}$ (where $f(\boldsymbol{x}) \in \mathscr{Y} \subset \mathcal{Y}$). The set S is called the Minimal Sufficient Explanation. Indeed, we have $CDP_S(\mathscr{Y}^\star; \boldsymbol{x}) = SDP_{\bar{S}}(\mathscr{Y}^\star; \boldsymbol{x})$. The computation of these probabilities is challenging and discussed in Section 4. We now focus on the minimal subset of features $S$ such that the model makes the desired decision with a given probability $\pi$.

**Definition 3.2. ( Minimal Divergent Explanations).** Given an instance $\boldsymbol{x}$ and a desired target $\mathscr{Y}^\star$, $S$ is a Divergent Explanation for probability $\pi > 0$, if $CDP_S(\mathscr{Y}^\star; \boldsymbol{x}) \geq \pi$, and no subset $Z$ of $S$ satisfies $CDP_Z(\mathscr{Y}^\star; \boldsymbol{x}) \geq \pi$. Hence, a Minimal Divergent Explanation is a Divergent Explanation with minimal size.

The set minimizing this probability is not unique, and we can have several Minimal Divergent Explanations. Note that the probability $\pi$ represents the minimum level required for a set to be chosen for generating counterfactuals, and its value should be as high as possible and depends on the use case. We have now enough material to define our main criterion for building a Local Counterfactual Rule (Local-CR):

**Definition 3.3. (Local Counterfactual Rule).** Given an instance $\boldsymbol{x}$, a desired target $\mathscr{Y}^\star \not\ni f(\boldsymbol{x})$, a Minimal Divergent Explanation $S$, the rectangle $C_S(\boldsymbol{x}; \mathscr{Y}^\star) = \prod_{i \in S}[a_i, b_i], a_i, b_i \in \overline{\mathbb{R}}$ is a Local Counterfactual Rule with probability $\pi_C$ if

$$CRP_S(\mathscr{Y}^\star, \boldsymbol{x}, C_S(\boldsymbol{x}; y^\star)) \triangleq P(f(\boldsymbol{X}) \in \mathscr{Y}^\star | \boldsymbol{X}_S \in C_S(\boldsymbol{x}; \mathscr{Y}^\star), \boldsymbol{X}_{\bar{S}} = \boldsymbol{x}_{\bar{S}}) \geq \pi_C. \quad (3.1)$$

The $CRP_S$ is the Counterfactual Rule Probability.

The higher the probability $\pi_C$ is, the better the relevance of the rule $C_S(\boldsymbol{x}; \mathscr{Y}^\star)$ is, for this instance. Given a set $S$, we seek for the maximal rectangle in the direction $S$ satisfying Definition 3.1.

In practice, we can observe that the Local-CR $C_S(\cdot; \mathscr{Y}^\star)$ for neighbors $\boldsymbol{x}, \boldsymbol{x}'$ are often quite close, because the Minimal Divergent Explanations are similar and the corresponding rectangles often overlaps.

Hence, this motivates a generalisation of these Local-CR to hyperrectangle $\boldsymbol{R} = \prod_{i=1}^{d}[a_i, b_i], a_i, b_i \in \overline{\mathbb{R}}$ regrouping similar observations. We denote $\text{supp}(\boldsymbol{R}) = \{i : [a_i, b_i] \neq \overline{\mathbb{R}}\}$ the support of the rectangle, and we extend the Local-CR to Regional Counterfactual Rules (Regional-CR). In order to do it, we denote $\boldsymbol{R}_{\bar{S}} = \prod_{i \in \bar{S}}[a_i, b_i]$ as the rectangle with intervals of $\boldsymbol{R}$ in $\text{supp}(\boldsymbol{R}) \cap \bar{S}$ and we also defines the corresponding Counterfactual Decision Probability CDP (Definition 3.1) for rule $\boldsymbol{R}$ and subset $S$ as $CDP_S(\mathscr{Y}^{\star}; \boldsymbol{R}) = P(f(\boldsymbol{X}) \in \mathscr{Y}^{\star} | \boldsymbol{X}_{\bar{S}} \in \boldsymbol{R}_{\bar{S}})$. Therefore, we can also compute the Minimal Divergent Explanation for rule $\boldsymbol{R}$ using Definition 3.2 with the CDP for rules.

**Definition 3.4.** (**Regional Counterfactual Rule**). Given any rectangle $\boldsymbol{R}$, a desired target $\mathscr{Y}^{\star}$, a Minimal Divergent Explanation $S$ of $R$, the rectangle $C_S(\boldsymbol{R}; y^{\star}) = \prod_{i \in S}[a_i, b_i]$ is a Regional Counterfactual Rule with probability $\pi_C$ if

$$CRP_S(\mathscr{Y}^{\star}; \boldsymbol{R}, C_S(\boldsymbol{R}, \mathscr{Y}^{\star})) \triangleq P(f(\boldsymbol{X}) \in \mathscr{Y}^{\star} | \boldsymbol{X}_S \in C_S(\boldsymbol{R}, \mathscr{Y}^{\star}), \boldsymbol{X}_{\bar{S}} \in \boldsymbol{R}_{\bar{S}}) \geq \pi_C. \quad (3.2)$$

$CRP_S(\mathscr{Y}^{\star}; \boldsymbol{R}, C_S(\boldsymbol{R}))$ is the corresponding Counterfactual Rule Probability for rule $\boldsymbol{R}$.

**Remarks:** Local-CR and regional-CR differ slightly: for local, we condition by $\boldsymbol{X}_{\bar{S}} = \boldsymbol{x}_{\bar{S}}$ in Eq. 3.1, while for regional, we condition by $\boldsymbol{X}_{\bar{S}} \in \boldsymbol{R}_{\bar{S}}$. For computing regional-CR, we can start for a rectangle generated by any method, such as [Wang et al., 2017, Lin et al., 2020]. The only condition is that it contains a homogeneous group, i.e. with almost the same output. However, by default we use as initial rules the Sufficient Rules derived in [Amoukou and Brunel, 2021] as it handles regression problem. The Sufficient Rules are minimal support rectangles define for a given output $\mathscr{Y}$ as $C_S(\mathscr{Y}) = \Pi_{i \in S}[a_i, b_i]$ such that $\forall \boldsymbol{x} \in \mathcal{X}, \boldsymbol{x}_S \in C_S(\mathscr{Y}), P(f(\boldsymbol{X}) \in \mathscr{Y} | \boldsymbol{X}_S = \boldsymbol{x}_S) \geq \pi$.

# 4   Estimation of the $CDP$ and $CRP$

In order to compute the probabilities $CDP_S$ and $CRP_S$ for any $S$, we use a dedicated Random Forest (RF) $m_{k,n}$ that learns the model $f$ to explain. Indeed, the conditional probabilities $CDP_S$ and $CRP_S$ can be easily computed from a RF by combining the Projected Forest algorithm [Bénard et al., 2021a] and the Quantile Regression Forest [Meinshausen and Ridgeway, 2006]: hence we can estimate consistently the probabilities $CDP_S(\mathscr{Y}^{\star}; \boldsymbol{x})$. We adapt the approach used in [Amoukou and Brunel, 2021] and remind for the sake of completeness, the computation of the estimate of $SDP_S$.

## 4.1   Projected Forest and $CDP_S$

The estimator of the $SDP_S$ is built upon a learned Random Forest [Breiman et al., 1984]. A Random Forest (RF) is a predictor consisting of a collection of $k$ randomized trees (see [Loh, 2011] for a detailed description of decision tree). For each instance $\boldsymbol{x}$, the predicted value of the $j$-th tree is denoted $m_n(\boldsymbol{x}, \Theta_j)$ where $\Theta_j$ represents the resampling data mechanism in the $j$-th tree and the successive random splitting directions. The trees are then averaged to give the prediction of the forest as:

$$m_{k,n}(\boldsymbol{x}, \Theta_{1:k}, \mathcal{D}_n) = \frac{1}{k}\sum_{l=1}^{k} m_n(\boldsymbol{x}; \Theta_l, \mathcal{D}_n) \quad (4.1)$$

However, the RF can also be view as an adaptive nearest neighbor predictor. For every instance $\boldsymbol{x}$, the observations in $\mathcal{D}_n$ are weighted by $w_{n,i}(\boldsymbol{x}; \Theta_{1:k}, \mathcal{D}_n), i = 1, \ldots, n$. Therefore, the prediction of RF can be rewritten as

$$m_{k,n}(\boldsymbol{x}, \Theta_{1:k}, \mathcal{D}_n) = \sum_{i=1}^{n} w_{n,i}(\boldsymbol{x}; \Theta_{1:k}, \mathcal{D}_n)Y_i.$$

This emphasizes the central role played by the weights in the RF's algorithm, see [Meinshausen and Ridgeway, 2006, Amoukou and Brunel, 2021] for detailed description of the weights. Therefore, it naturally gives estimators of other quantities e.g., Cumulative hazard function [Ishwaran et al., 2008], Treatment effect [Wager and Athey, 2017], conditional density [Du et al., 2021]. For instance, Meinshausen and Ridgeway [2006] showed that we can used the same weights to estimate the Conditional Distribution Function with the following estimator:

$$\widehat{F}(y|\boldsymbol{X} = \boldsymbol{x}, \Theta_{1:k}, \mathcal{D}_n) = \sum_{i=1}^{n} w_{n,i}(\boldsymbol{x}; \Theta_{1:k}, \mathcal{D}_n)\mathbb{1}_{Y_i \leq y} \quad (4.2)$$

In another direction, Bénard et al. [2021a] introduced the Projected Forest algorithm [Bénard et al., 2021c,a] that aims to estimate $E[Y|\boldsymbol{X}_S]$ by modifying the RF's prediction algorithm.

**Projected Forest:** To estimate $E[Y|\boldsymbol{X}_S = \boldsymbol{x}_S]$ instead of $E[Y|\boldsymbol{X} = \boldsymbol{x}]$ using a RF, Bénard et al. [2021b] suggests to simply ignore the splits based on the variables not contained in $S$ from the tree predictions. More formally, it consists of projecting the partition of each tree of the forest on the subspace spanned by the variables in S. The authors also introduced an algorithmic trick that computes the projected partition efficiently without modifying the initial tree structures. We drop observations down in the initial trees, ignoring the splits which use a variable not in $S$: when a split involving a variable outside of $S$ is met, the observations are sent both to the left and right children nodes. Therefore, each instance falls in multiple terminal leaves of the tree. We drop the new query point $\boldsymbol{x}_S$ down the tree, following the same procedure, and gather the set of terminal leaves where $\boldsymbol{x}_S$ falls. Next, we collect the training observations which belong to every terminal leaf of this collection, in other words, we keep only the observations that fall in the intersection of the leaves where $\boldsymbol{x}_S$ falls. Finally, we average the outputs $Y_i$ of the selected training points to generate the estimation of $E[Y|\boldsymbol{X}_S = \boldsymbol{x}_S]$. Notice that this algorithm converges asymptotically to the true projected conditional expectation $E[Y|\boldsymbol{X}_S = \boldsymbol{x}_S]$.

As the RF, the PRF gives also a weight to each observation. The associated PRF is denoted $m_{k,n}^{(\boldsymbol{x}_S)}(\boldsymbol{x}_S) = \sum_{i=1}^{n} w_{n,i}(\boldsymbol{x}_S)Y_i$. Therefore, as the weights of the original forest was used to estimate the CDF in equation 4.2, Amoukou and Brunel [2021] used the weights of the Projected Forest Algorithm to estimate the $SDP$ as $\widehat{SDP}_S(\mathscr{Y};\boldsymbol{x}) = \sum_{i=1}^{n} w_{n,i}(\boldsymbol{x}_S)\mathbb{1}_{Y_i \in \mathscr{Y}}$. The idea is essentially to replace $Y_i$ by $\mathbb{1}_{Y_i \in \mathscr{Y}}$ in the Projected Forest equation defined above. The authors also show that this estimator converges asymptotically to the true $SDP_S$. Therefore, we can estimate the $CDP$ with the following estimator

$$\widehat{CDP}_S(\mathscr{Y}^{\star};\boldsymbol{x}) = \sum_{i=1}^{n} w_{n,i}(\boldsymbol{x}_{\bar{S}})\mathbb{1}_{Y_i \in \mathscr{Y}^{\star}}. \tag{4.3}$$

**Remarks:** Note that we only give the estimator of the $CDP_S$ of an instance $\boldsymbol{x}$. The estimator of the $CDP_S$ of a rule $R$ will be discussed in the next section as it is related to the estimator of the $CRP_S$.

## 4.2 Regional RF and $CRP_S$

In this section, we focus on the estimation of the $CRP_S(\mathscr{Y}^{\star}, \boldsymbol{x}, C_S(\boldsymbol{x};\mathscr{Y}^{\star})) = P(f(\boldsymbol{X}) \in \mathscr{Y}^{\star}|\boldsymbol{X}_S \in C_S(\boldsymbol{x};\mathscr{Y}^{\star}), \boldsymbol{X}_{\bar{S}} = \boldsymbol{x}_{\bar{S}})$ and $CRP_S(\mathscr{Y}^{\star}, \boldsymbol{R}, C_S(\boldsymbol{R};\mathscr{Y}^{\star})) = P(f(\boldsymbol{X}) \in \mathscr{Y}^{\star}|\boldsymbol{X}_S \in C_S(\boldsymbol{R};\mathscr{Y}^{\star}), \boldsymbol{X}_{\bar{S}} \in \boldsymbol{R}_{\bar{S}})$. For simplicity, we remove the dependency of the rectangles in $\mathscr{Y}^{\star}$. Based on the previous Section, we already know that the estimators using the RF will be in the form of $\widehat{CRP}_S(\mathscr{Y}^{\star}, \boldsymbol{x}, C_S(\boldsymbol{x})) = \sum_{i=1}^{n} w_{n,i}(\boldsymbol{x})\mathbb{1}_{Y_i \in \mathscr{Y}^{\star}}$, thus we only need to find the right weighting. The main challenge is that we have a condition based on a region, e.g., $\boldsymbol{X}_S \in C_S(\boldsymbol{x})$ or $\boldsymbol{X}_{\bar{S}} \in \boldsymbol{R}_{\bar{S}}$ (regional-based) instead of condition of type $\boldsymbol{X}_S = \boldsymbol{x}_S$ (fixed value-based) as usually. However, we introduced a natural generalization of the RF algorithm to make predictions when the conditions are both regional-based and fixed value-based. Thus, the case where there are only regional-based conditions are naturally derived.

**Regional RF to estimate** $CRP_S(\mathscr{Y}^{\star}, \boldsymbol{x}, C_S(\boldsymbol{x})) = P(f(\boldsymbol{X}) \in \mathscr{Y}^{\star}|\boldsymbol{X}_S \in C_S(\boldsymbol{x}), \boldsymbol{X}_{\bar{S}} = \boldsymbol{x}_{\bar{S}})$: The algorithm is based on a slight modification of RF. Its works as follow: we drop the observations in the initial trees, if a split used variable $i \in \bar{S}$, i.e., fixed value-based condition, we use the classic rules of RF, if $x_i \leq t$, the observations go to the left children, otherwise the right children. However, if a split used variable $i \in S$, i.e, regional-based condition, we use the rectangles $C_S(\boldsymbol{x}) = \prod_{i=1}^{|S|}[a_i, b_i]$. The observations are sent to the left children if $b_i \leq t$, right children if $a_i > t$ and if $t \in [a_i, b_i]$ the observations are sent both to the left and right children. Therefore, we use the weights of the Regional RF algorithm to estimate the $CRP_S$ as in equation 4.3, the estimator is $\widehat{CRP}_S(y^{\star}; \boldsymbol{x}, C_S(\boldsymbol{x})) = \sum_{i=1}^{n} w_{n,i}(\boldsymbol{x})\mathbb{1}_{Y_i = y^{\star}}$. A more detailed version of the algorithm is provided and discussed in Appendix.

To estimate the $CDP$ of a rule $CDP_S(\mathscr{Y}^{\star}; \boldsymbol{R}) = P(f(\boldsymbol{X}) \in \mathscr{Y}^{\star}|\boldsymbol{X}_{\bar{S}} \in \boldsymbol{R}_{\bar{S}})$, we just have to apply the projected Forest algorithm to the Regional RF, i.e., when a split involving a variable outside of $\bar{S}$ is met, the observations are sent both to the left and right children nodes, otherwise we use the Regional RF split rule, i.e., if an interval of $\boldsymbol{R}_{\bar{S}}$ is below $t$, the observations go to the left children, otherwise the right children and if $t$ is in the interval, the observations go to the left and right children.

271 The estimator of the $CRP_S(\mathscr{Y}^\star; \boldsymbol{R}, C_S(\boldsymbol{R}))$ for rule is also derived from the Regional RF. Indeed, it
272 is a special case of the Regional RF algorithm where there are only regional-based conditions.

# 5   Learning the Counterfactual Rules

274 We compute the Local and Regional CR using the estimators of the previous section. First, we find
275 the Minimal Divergent Explanation in the same way as Minimal Sufficient Explanation can be found
276 [Amoukou and Brunel, 2021]. As the exploration of all possible subsets is exponential, we search
277 the Minimal Divergent Subset among the $K = 10$ most frequently selected variables in the RF $m_{k,n}$
278 used to estimate the probabilities $CDP_S, CRP_S$ ($K$ is an hyper-parameter to select according to the
279 use case and computational power). We can also use any importance measure.

280 Given an instance $\boldsymbol{x}$ or rectangle $\boldsymbol{R}$ (and set $\mathscr{Y}^\star$) and their corresponding Minimal Divergent
281 Explanation S, we want to find a rule $C_S(\boldsymbol{x}) = \prod_{i \in S}[a_i, b_i]$ s.t. given $\boldsymbol{X}_{\bar{S}} = \boldsymbol{x}_{\bar{S}}$ or $\boldsymbol{X}_{\bar{S}} \in \boldsymbol{R}_{\bar{S}}$ and
282 $\boldsymbol{X}_S \in C_S(\boldsymbol{x})$, the probability that $Y \in \mathscr{Y}^\star$ is high. More formally, we want: $P(f(\boldsymbol{X}) \in \mathscr{Y}^\star | \boldsymbol{X}_S \in$
283 $C_S(\boldsymbol{x}), \boldsymbol{X}_{\bar{S}} = \boldsymbol{x}_{\bar{S}})$ or $P(f(\boldsymbol{X}) \in \mathscr{Y}^\star | \boldsymbol{X}_S \in C_S(\boldsymbol{x}), \boldsymbol{X}_{\bar{S}} \in \boldsymbol{R}_{\bar{S}})$ above $\pi_C$.

284 The computation of the rectangles $C_S(\boldsymbol{x}) = \prod_{i \in S|}[a_i, b_i]$ relies heavily on our use of RF and on the
285 algorithmic trick of the projected RF. Indeed, the rectangles defining the rules arise naturally from RF,
286 while AReS [Rawal and Lakkaraju, 2020] relies on binned variables to generate candidate rules and
287 tests all these possible rules for choosing an optimal one. We overcome the computational burden
288 and the challenge of choosing the number of bins.

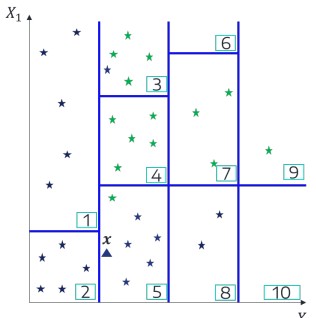 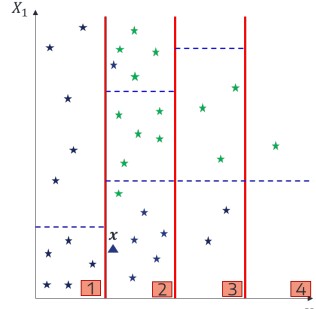 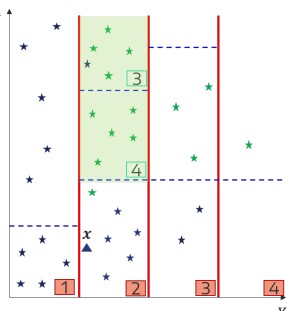

Figure 2: The partition of the RF learned to classify the toy data (Green/Blue stars). Its has 10 leaves. The explainee $\boldsymbol{x}$ is the Blue triangle in leaf 5.

Figure 3: The partition of the projected Forest when we condition on $X_0$, i.e., ignoring the splits based on $X_1$ (the dashed lines).

Figure 4: The optimal CR for $\boldsymbol{x}$ when we condition given $X_0 = x_0$ is the Green region, its corresponds to the union of leaf 3 and 4 of the forest

289 To illustrate the idea, we use a two-dimensional data $(X_0, X_1)$ with label Y represented as Green/Blue
290 stars in figure 2. We fit a Random Forest to classify this dataset and show its partition in figure 2. The
291 explainee $\boldsymbol{x}$ is the Blue triangle observation.

292 By looking at the different cells/leaves of the RF, we can guess that the Minimal Divergent Explanation
293 of $\boldsymbol{x}$ is $S = X_1$. Indeed, in figure 3, we observe the leaves of the Projected Forest when we do not
294 condition on $S = X_1$, thus projected the RF's partition only on the subspace $X_0$. Its consists of
295 ignoring all the splits in the other directions (here the $X_1$-axis), thus $\boldsymbol{x}$ falls in the projected leaf 2
296 (see figure 3) and its $CDP$ is $CDP_{X_1}(\text{Green}; \boldsymbol{x}) = \frac{10\,\text{Green}}{10\,\text{Green}+17\,\text{Blue}} = 0.58$.

297 Finally, the problem of finding the optimal rectangle $C_S(\boldsymbol{x}) = [a_i, b_i]$ in the direction of $X_1$ s.t. the
298 decision changes can be easily solved by using the leaves of the RF. In fact, by looking at the leaves
299 of the RF (figure 2) of the observations that belong in the projected RF leaf 2 (figure 3) where $\boldsymbol{x}$ falls,
300 we see in figure 4 that the optimal rectangle to change the decision given $X_0 = x_0$ or being in the
301 projected RF leaf 2 is the union of the intervals on $X_1$ of the leaf 3 and 4 of the RF (see the Green
302 region of figure 4).

303 Given an instance $\boldsymbol{x}$ and its Minimal Divergent Explanation $S$, the first step is the collect of the
304 observations which belong to the leaf of the Projected Forest given $\bar{S}$ where $\boldsymbol{x}$ falls. It corre-
305 sponds to the observations that has positive weights in the computation of the $CDP_S(\mathscr{Y}^\star; \boldsymbol{x}) =$

$\sum_{i=1}^{n} w_{n,i}(\boldsymbol{x}_{\bar{S}})\mathbb{1}_{Y_i \in \mathscr{Y}^{\star}}$, i.e., $\{\boldsymbol{x}_i : w_{n,i}(\boldsymbol{x}_{\bar{S}}) > 0\}$. Then, we used the partition of the original forest to find the possible leaves $C_S(\boldsymbol{x})$ in the direction $S$. The possible leaves is among the RF's leaves of the collected observations $\{\boldsymbol{x}_i : w_{n,i}(\boldsymbol{x}_{\bar{s}}) > 0\}$. Let denote $L(\boldsymbol{x}_i)$ the leaves of the observations $\boldsymbol{x}_i$ with $w_{n,i}(\boldsymbol{x}_{\bar{S}}) > 0$. A possible leaf is a leaf $L(\boldsymbol{x}_i)$ s.t. $CRP_S(\mathscr{Y}^{\star}, \boldsymbol{x}, L(\boldsymbol{x}_i)_S) = P(f(\boldsymbol{X}) \in \mathscr{Y}^{\star}|\boldsymbol{X}_S \in L(\boldsymbol{x}_i)_S, \boldsymbol{X}_{\bar{S}} = \boldsymbol{x}_{\bar{S}}) \geq \pi_C$. Finally, we merge all the neighboring possible leaves to get the largest rectangle, and this maximal rectangle is the counterfactual rule. Note that the union of the possible leaves is not necessary a connected space, thus we can have multiple counterfactual rules.

We apply the same idea to find the regional CR. Given a rule $\boldsymbol{R}$ and its Minimal Divergent Explanation $S$, we used the Projection given $\boldsymbol{X}_{\bar{S}} \in \boldsymbol{R}_{\bar{S}}$ to find the compatible observations and their leaves and combine the possible ones to obtain the regional CR that has $CRP_S(\mathscr{Y}^{\star}, \boldsymbol{R}, C_S(\boldsymbol{R})) \geq \pi_C$. For example, if we consider the leaf 5 of the original forest as a rule: If $\boldsymbol{X} \in$ Leaf 5, then predict Blue. Its Minimal Divergent Explanation is also $S = X_1$. The R-CR would also be the Green region in figure 4. Indeed, if we satisfy the $X_0$ condition of the leaf 5 and $X_1$ condition of the leaf 3 and 4, then the decision change to Green.

# 6 Experiments

To demonstrate the performance of our framework, we conduct two experiments on real-world datasets. The first consists of showing how we can use the *Local Counterfactual Rules* for explaining a regression model. In the second experiment, we compare our approaches with the 2 baselines methods in classification problem: (1) **CET** [Kanamori et al., 2022], which partition the input space using a decision tree and associate a vector perturbation for each leaf, (2) **AReS** [Rawal and Lakkaraju, 2020] performs an exhaustive search for finding global counterfactual rules, but we used the implementation of Kanamori et al. [2022] that adapts the algorithm for returning counterfactuals samples instead of rules. We compare the methods only in classification problem as most prior works do not deal regression problem. In all experiments, we split our dataset into train (75%) - test (25%), and we learn a model $f$, a LightGBM *(estimators=50, nb leaves=8)*, on the train set that is the explainee. We learn $f$'s predictions on the train set with an approximating RF $m_{nb,n}$ *(estimators=20, max depth=10)*: **that** will be used to generate the CR with $\pi = 0.9$. The used parameters for **AReS**, **CET** are *max rules=8, bins=10* and *max iterations=1000, max leaf=8, bins=10* respectively. Due to page limitation, the detailed parameters of each method are provided in Appendix.

**Sampling CE using the Counterfactual Rules:** Notice that our approaches cannot be directly compare with the baseline methods since they all return counterfactual samples while we give rules (range of vector values) that permit to change the decision with high probability. However, we adapt the CR to generate also counterfactual samples using a generative model. For example, given an instance $\boldsymbol{x} = (\boldsymbol{x}_S, \boldsymbol{x}_{\bar{S}})$, target $\mathscr{Y}^{\star}$ and its counterfactual rule $C_S(\boldsymbol{x}; \mathscr{Y}^{\star})$, we want to find a sample $x^{\star} = (\boldsymbol{z}_S, \boldsymbol{x}_{\bar{S}})$ with $\boldsymbol{z}_S \in C_S(\boldsymbol{x}, \mathscr{Y}^{\star})$ s.t $x^{\star}$ is an in-distribution sample and $f(x^{\star}) \in \mathscr{Y}^{\star}$. Instead of using a complex conditional generative model as [Xu et al., 2019, Patki et al., 2016] that can be difficult to calibrate, we use an energy-based generative approach [Grathwohl et al., 2020, Lecun et al., 2006]. The core idea is to find $\boldsymbol{z}_S \in C_S(\boldsymbol{x}, y^{\star})$ s.t. $x^{\star}$ maximize a given energy score to ensure that it is an in-distribution sample. As an example of an energy function, we use the negative outlier score of an Isolation Forest [Liu et al., 2008]. We use Simulated Annealing (see [Guilmeau et al., 2021] for a review) to maximize the negative outlier score using the information of the counterfactual rules $C_S(\boldsymbol{x}; \mathscr{Y}^{\star})$. In fact, the range values given by the CR $C_S(\boldsymbol{x}; \mathscr{Y}^{\star})$ reduce the search space for $\boldsymbol{z}_S$ drastically. We used the training set $\mathcal{D}_n$ to find the possible values i.e., we defined $P_i, P_S$ as the list of values of the variable $i \in S$ found in $\mathcal{D}_n$ and $P_S = \{\boldsymbol{z}_S = (z_1, \ldots, z_S) : \boldsymbol{z}_S \in C_S(\boldsymbol{x}, y^{\star}), z_i \in P_i\}$ the possible values of $\boldsymbol{z}_S$ respectively. Then, we sample $\boldsymbol{z}_S$ in the set $P_S$ and use Simulated Annealing to find a $\boldsymbol{x}^{\star}$ that maximizes the negative outlier score. Note that the algorithm is the same for sampling CE with the Regional-CR. A more detailed version of the algorithm is provided in Appendix.

Finally, we compare the methods on unseen observations using three criteria. *Correctness* is the average number of instances for which acting as prescribed change to the desired prediction. *Plausibility* is the average number of inlier (predict by an Isolation Forest) in the counterfactual samples. *Sparsity* is the average number of features that have been changed, and especially for the global counterfactual methods (AReS, Regional-CR) that do not ensure to cover all the instances, we compute *Coverage* that corresponds to the average number of unseen observations we cover.

**Local counterfactual rules for regression:** We give recourse for the **California House Price**
dataset [Kelley Pace and Barry, 1997] derived from the 1990 U.S. census. We have information about
each district (demography, . . . ), and the goal is to predict the median house value of each district.

To illustrate the efficiency of the Local-CR, we select all the observations in the test set having a price
lower than $100k$ (1566 houses), and we aim to find the recourse that permit to increase their price
: we want the price $y$ to be in the interval $\mathcal{Y}^\star = [200k, 250k]$. For each instance $x$, we compute
the Minimal Divergent Explanation $S$, the Local-CR $C_S(x; [200k, 250k])$ and a CE using the
Simulated Annealing as described above. We succeed in changing the decision of all the observations
(*Correctness* $= 1$) and most of them passed the outlier test with *Plausibility* $= 0.92$. On top of that,
our Local-CR have sparse support (*Sparsity* $= 4.45$). For example, the Local-CR of the instance $x =$
(Longitude=-118.2, latitude=33.8, housing median age=26, total rooms=703,
total bedrooms=202, population=757, households=212, median income=2.52) is
$C_S(x, [200k, 250k])$ = (total room $\in$ [2132, 3546], total bedrooms $\in$ [214, 491]). It
means if total room and total bedrooms satisfy the conditions in $C_S(x, [200k, 250k])$ and
the remaining features of $x$ is fixed, then the probability that the price is in $[200k, 250k]$ is 0.97.

**Comparisons of Local-CR and Regional-CR with baselines (AReS, CET):** We use 3 real-world
datasets: **Diabetes** [Kaggle, 2016] contains diagnostic measurements and aims to predict whether
or not a patient has diabetes, **Breast Cancer Wisconsin (BCW)** [Dua and Graff, 2017] consists of
predicting if a tumor is benign or not using the characteristic of the cell nuclei, and **Compas** [Larson
et al., 2016] was used to predict recidivism, and it contains information about the criminal history,
demographic attributes. During the evaluation, we observe that **AReS, CET** are very sensitive to the
number of bins and the maximal number of rules or actions as noticed by [Ley et al., 2022]. A bad
parameterization gives completely useless explanations. Moreover, a different model needs to be
trained for each class to be accurate, while we only need to have a RF that has good precision.

In table 1, we notice that the Local and Regional-CR succeed in changing decisions with a high
accuracy in all datasets, outperforming **AReS** and **CET** with a large margin on **BCW**, and **Diabetes**.
Moreover, we notice that the baselines struggle to change at the same time the positive and negative
class, (e.g. CET has *Acc*=1 in the positive class, and 0.21 for the negative class on **BCW**) or when
they have a good *Acc*, the CE are not plausible. For instance, CET has *Acc*=0.98 and *Psb*=0 on
**Compas**, meaning that all the CE are outlier. Regarding the coverage of the global CE, CET covers
all the instances as it partitions the space, but we observe that **AReS** has a smaller *Coverage*=
$\{0.43, 0.44, 0.81\}$ than the Regional-CR which has $\{1, 0.7, 1\}$ for **BCW, Diabetes, and Compas**
respectively. To sum up, the CR is easier to train and provides more accurate and plausible rules than
the baselines methods.

Table 1: Results of the *Correctness* (Acc), *Plausibility*, and *Sparsity* (Sprs) of the different methods.
We compute each metric according to the positive (Pos) and negative (Neg) class.

| | COMPAS | | | | | | BCW | | | | | | Diabetes | | | | | |
|---|---|---|---|---|---|---|---|---|---|---|---|---|---|---|---|---|---|---|
| | Acc | | Psb | | Sps | | Acc | | Psb | | Sps | | Acc | | Psb | | Sps | |
| | Pos | Neg | Pos | Neg | Pos | Neg | Pos | Neg | Pos | Neg | Pos | Neg | Pos | Neg | Pos | Neg | Pos | Neg |
| **L-CR** | 1 | 0.9 | 0.87 | 0.73 | 2 | 4 | 1 | 1 | 0.96 | 1 | 9 | 7 | 0.97 | 1 | 0.99 | 0.8 | 3 | 4 |
| **R-CR** | 0.9 | 0.98 | 0.74 | 0.93 | 2 | 3 | 0.89 | 0.9 | 0.94 | 0.93 | 9 | 9 | 0.99 | 0.99 | 0.9 | 0.87 | 3 | 4 |
| **AReS** | 0.98 | 1 | 0.8 | 0.61 | 1 | 1 | 0.63 | 0.34 | 0.83 | 0.80 | 4 | 3 | 0.73 | 0.60 | 0.77 | 0.86 | 1 | 1 |
| **CET** | 0.85 | 0.98 | 0.7 | 0 | 2 | 2 | 1 | 0.21 | 0.6 | 0.80 | 8 | 2 | 0.84 | 1 | 0.60 | 0.20 | 6 | 6 |

# 7 Conclusion

Most current works that generate CE are implicit through an optimization process or a brunch of
random samples, thus lacking guarantees. For this reason, we rethink CE as *Counterfactual Rules*.
For any individual or sub-population, it gives the simplest policies that change the decision with
high probability. Our approach learns robust, plausible, and sparse adversarial regions where the
observations should be moved. We make central use of Random Forests, which give consistent
estimates of the interest probabilities and naturally give the counterfactual rules we want to extract.
In addition, it permits us to deal with regression problems and continuous features. Consequently,
our methods are suitable for all datasets where tree-based model performs well (e.g., tabular data). A
prospective work is to evaluate the robustness of our methods to noisy human responses, i.e., when
the prescribed recourse is not implemented exactly, and to refine the methodology for selecting the
threshold probabilities $\pi$ and $\pi_C$.

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
