# Supplementary materials: Rethinking Counterfactual Explanations as Local and Regional Counterfactual Policies

## 1 Contents

Submitted to 36th Conference on Neural Information Processing Systems (NeurIPS 2022). Do not distribute.

## A  Regional RF detailed

In this section, we give a simple application of the Regional RF algorithm to better understand how it works. Recall that the regional RF is a generalization of the RF's algorithm to give prediction even when we condition given a region, e.g., to estimate $E(f(\boldsymbol{X}) | \boldsymbol{X}_S \in C_S(\boldsymbol{x}), \boldsymbol{X}_{\bar{S}} = \boldsymbol{x}_{\bar{S}})$ with $C_S(\boldsymbol{x}) = \prod_{i=1}^{|S|}[a_i, b_i], a_i, b_i \in \bar{\mathbb{R}}$ a hyperrectangle. The algorithm works as follows: we drop the observations in the initial trees, if a split used variable $i \in \bar{S}$, a fixed value-based condition, we used the classic rules i.e., if $x_i \leq t$, the observations go to the left children, otherwise the right children. However, if a split used variable $i \in S$, regional-based condition, we used the hyperrectangle $C_S(\boldsymbol{x}) = \prod_{i=1}^{|S|}[a_i, b_i]$. The observations are sent to the left children if $b_i \leq t$, right children if $a_i > t$ and if $t \in [a_i, b_i]$ the observations are sent both to the left and right children.

To illustrate how it works, we use a two dimensional variables $\boldsymbol{X} \in \mathbb{R}^2$, a simple decision tree $f$ represented in figure 1, and want to compute for $\boldsymbol{x} = [1.5, 1.9]$, $E(f(\boldsymbol{X})|X_1 \in [2, \ 3.5], \boldsymbol{X}_0 = 1.5)$. We assume that $P(X_1 \in [2, \ 3.5] | X_0 = 1.5) > 0$ and denoted $T_1$ as the set of the values of the splits based on variables $X_1$ of the decision tree. One way of estimating this conditional mean is by using Monte Carlo sampling. Therefore, there are two cases :

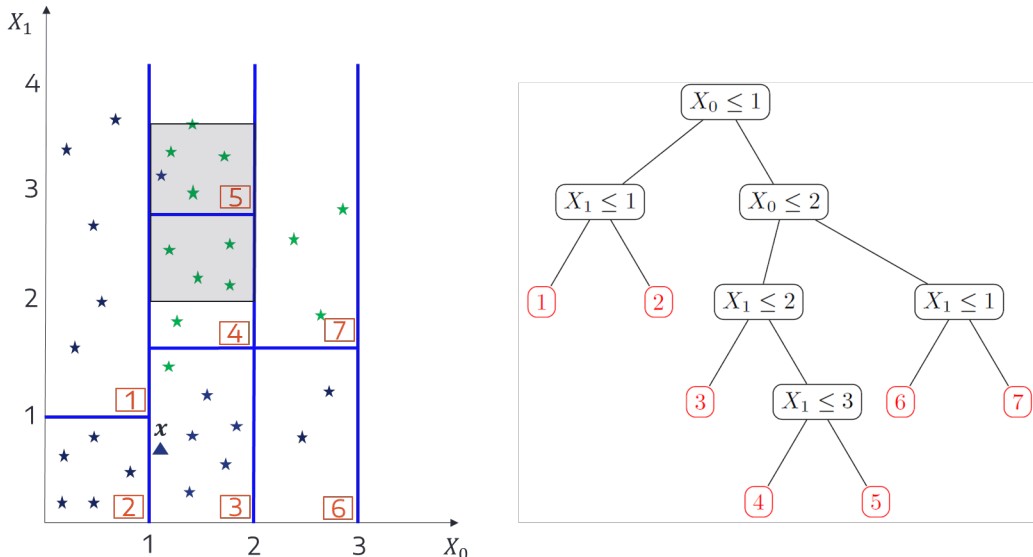

Figure 1: Representation of a simple decision tree (right figure) and its associated partition (left figure). The gray part in the partition corresponds to the region $[2, \ 3.5] \times [1, 2]$

- If $\forall t \in T_1$, $t \leq 2$ or $t > 3$, then all the observations sampled s.t. $\tilde{X}_i \sim \mathcal{L}(\boldsymbol{X} | X_1 \in [2, \ 3.5], \boldsymbol{X}_0 = 1.5)$ follow the same path and fall in the same leaf. The Monte Carlo estimator of the decision tree $E(f(\boldsymbol{X})|X_1 \in [2, \ 3.5], \boldsymbol{X}_0 = 1.5)$ is equal to the output of the Regional RF algorithm.
    - For instance, a special case of the case above is: if $\forall t \in T_1, t \leq 2$, and we sample using $\mathcal{L}(\boldsymbol{X} | X_1 \in [2, \ 3.5], \boldsymbol{X}_0 = 1.5)$, then all the observations go to the right children when they encounters a node using $X_1$ and fall in the same leaf.

- If $\exists \ t \in T_1$ and $t \in [2, \ 3.5]$, then the observations sampled s.t. $\tilde{X}_i \sim \mathcal{L}(\boldsymbol{X} | X_1 \in [2, \ 3.5], \boldsymbol{X}_0 = 1.5)$ can fall in multiple terminal leaf depending on if their coordinates $x_1$ is lower than $t$. Following our example, if we generate samples using $\mathcal{L}(\boldsymbol{X} | X_1 \in [2, \ 3.5], \boldsymbol{X}_0 = 1.5)$, the observations will fall in the gray region of figure 1, and thus can fall in node 4 or 5. Therefore, the true estimate is:

$$E(f(\boldsymbol{X})|X_1 \in [2, \ 3.5], \boldsymbol{X}_0 = 1.5)$$
$$= p(X_1 \leq 2.9 \, | X_0 = 1.5) * E[f(\boldsymbol{X}) \, | \boldsymbol{X} \in L_4] + p(X_1 > 2.9 \, | X_0 = 1.5) * E[f(\boldsymbol{X}) \, | \boldsymbol{X} \in L_5]$$
$$\text{(A.1)}$$

Concerning the last case ($t \in [2, \ 3.5]$), we need to estimate the different probabilities $p(X_1 \leq 2.9 \,|X_0 = 1.5), p(X_1 > 2.9 \,|X_0 = 1.5)$ to compute $E(f(\boldsymbol{X})|\boldsymbol{X}_1 \in [2, \ 3.5], \boldsymbol{X}_0 = 1.5)$, but these probabilities are difficult to estimate in practice. However, we argue that we can ignore these splits, and thus do no need to fragment the query region using the leaves of the tree. Indeed, as we are no longer interest in a point estimate but regional (population mean) we do not need to go to the level of the leaves. We propose to ignore the splits of the leaves that divide the query region. For instance, the leaves 4 and 5 split the region $[2, \ 3.5]$ in two cells, by ignoring these splits we estimate the mean of the gray region by taking the average output of the leaves 4 and 5 instead of computing the mean weighted by the probabilities as in Eq. A.1. Roughly, it consists to follow the classic rules of a decision tree (if the region is above or below a split) and ignore the splits that are in the query region, i.e., we average the output of all the leaves that are compatible with the condition $\boldsymbol{X}_1 \in [2, \ 3.5], \boldsymbol{X}_0 = 1.5$. We think that it leads to a better approximation for two reasons. First, we observe that the case where t is in the region and thus divides the query region does not happen often. Moreover, the leaves of the trees are very small in practice, and taking the mean of the observations that fall in the union of leaves that belong to the query region is more reasonable than computing the weighted mean and thus trying to estimate the different probabilities $p(X_1 \leq 2.9 \,|X_0 = 1.5), p(X_1 > 2.9 \,|X_0 = 1.5)$.

## B  Additional experiments

In table 1, we compare the *Correctness* (Acc), *Plausibility* (Psb), and *Sparsity* (Sprs) of the different methods on additonal real-world datasets: FICO [FICO, 2018], NHANESI [CDC, 1999-2022].

We observe that the L-CR, and R-CR outperform the baseline methods by a large margin on *Correctness* and *Plausibility*. The baseline methods still struggle to change at the same time the positive and negative class. In addition, AReS and CET give better sparsity, but their counterfactual samples are less plausible than the ones generated by the CR.

Table 1: Results of the *Correctness* (Acc), *Plausibility*, and *Sparsity* (Sprs) of the different methods. We compute each metric according to the positive (Pos) and negative (Neg) class.

| | FICO | | | | | | NHANESI | | | | | |
|---|---|---|---|---|---|---|---|---|---|---|---|---|
| | Acc | | Psb | | Sps | | Acc | | Psb | | Sps | |
| | Pos | Neg | Pos | Neg | Pos | Neg | Pos | Neg | Pos | Neg | Pos | Neg |
| **L-CR** | 0.98 | 0.94 | 0.98 | 0.99 | 5 | 5 | 0.99 | 0.98 | 0.98 | 0.97 | 5 | 6 |
| **R-CR** | 0.90 | 0.94 | 0.98 | 0.99 | 9 | 8.43 | 0.86 | 0.95 | 0.96 | 0.99 | 7 | 7 |
| **AReS** | 0.34 | 0.01 | 0.85 | 0.86 | 2 | 1 | 0.06 | 1 | 0.87 | 0.92 | 1 | 1 |
| **CET** | 0.76 | 0 | 0.76 | 0.60 | 2 | 2 | 0 | 0.40 | 0.82 | 0.56 | 0 | 5 |

## C  Simulated annealing to generate counterfactual samples using the Counterfactual Rules

```
import numpy as np

def generate_candidate(x, S, x_train, C_S, n_samples):
    """
    Generate sample by sampling marginally between the features value
    of the training observations.
    Args:
        x (numpy.ndarray)): 1-D array, an observation
        S (list): contains the indices of the variables on which to
    condition
        x_train (numpy.ndarray)): 2-D array represent the training
    samples
        C_S (numpy.ndarray)): 3-D (#variables x 2 x 1) representing
    the hyper-rectangle on which to condition
        n_samples (int): number of samples
    Returns:
        The generated samples
```

```python
    """
    x_poss = [x_train[(C_S[i, 0] <= x_train[:, i]) * (x_train[:, i] <=
    C_S[i, 1]), i] for i in S]
    x_cand = np.repeat(x.reshape(1, -1), repeats=n_samples, axis=0)

    for i in range(len(S)):
        rdm_id = np.random.randint(low=0, high=x_poss[i].shape[0],
    size=n_samples)
        x_cand[:, S[i]] = x_poss[i][rdm_id]

    return x_cand

def simulated_annealing(outlier_score, x, S, x_train, C_S, batch,
    max_iter, temp, max_iter_convergence):
    """
    Generate sample X s.t. X_S \in C_S using simulated annealing and
    outlier score.
    Args:
        outlier_score (lambda functon): outlier_score(X) return a
    outlier score. If the value are negative, then the observation is
    an outlier.
        x (numpy.ndarray)): 1-D array, an observation
        S (list): contains the indices of the variables on which to
    condition
        x_train (numpy.ndarray)): 2-D array represent the training
    samples
        C_S (numpy.ndarray)): 3-D (#variables x 2 x 1) representing
    the hyper-rectangle on which to condition
        batch (int): number of sample by iteration
        max_iter (int): number of iteration of the algorithm
        temp (double): the temperature of the simulated annealing
    algorithm
        max_iter_convergence (double): minimun number of iteration to
    stop the algorithm if it find an in-distribution observation

    Returns:
        The generated sample, and its outlier score
    """

    best = generate_candidate(x, S, x_train, C_S, n_samples=1)
    best_eval = outlier_score(best)[0]
    curr, curr_eval = best, best_eval

    it = 0
    for i in range(max_iter):

        x_cand = generate_candidate(curr, S, x_train, C_S, batch)
        score_candidates = outlier_score(x_cand)

        candidate_eval = np.max(score_candidates)
        candidate = x_cand[np.argmax(score_candidates)]

        if candidate_eval > best_eval:
            best, best_eval = candidate, candidate_eval
            it = 0
        else:
            it += 1

        # check convergence
        if best_eval > 0 and it > max_iter_convergence:
            break

        diff = candidate_eval - curr_eval
        t = temp / np.log(float(i + 1))
```

```
68        metropolis = np.exp(-diff / t)
69
70        if diff > 0 or rand() < metropolis:
71            curr, curr_eval = candidate, candidate_eval
72
73    return best, best_eval
```

Listing 1: The simulated annealing algorithm to generate samples that satisfy the condition CR

# D  Parameters detailed

In this section, we give the different parameters of each method. For all methods and datasets, we first used a greedy search given a set of parameters. For AReS, we use the following set of parameters:

- max rule = $\{4, 6, 8\}$, max rule length $= \{4, 8\}$, max change num $= \{2, 4, 6\}$,
- minimal support $= 0.05$, discretization bins $= \{10, 20\}$,
- $\lambda_{acc} = \lambda_{cov} = \lambda_{cst} = 1$.

For CET, we search in the following set of parameters:

- max iterations $= \{500, 1000\}$,
- max leaf size $= \{4, 6, 8, -1\}$,
- $\lambda = 0.01, \gamma = 1$.

Finally, for the Counterfactual Rules, we used the following parameters:

- nb estimators $= \{20, 50\}$, max depth$= \{8, 10, 12\}$,
- $\pi = 0.9$, $\pi_C = 0.9$.

We obtained the same optimal parameters for all datasets:

- AReS: max rule $= 4$, max rule length$= 4$, max change num $= 4$, minimal support $= 0.05$, discretization bins $= 10$, $\lambda_{acc} = \lambda_{cov} = \lambda_{cst} = 1$
- CET: max iterations $= 1000$, max leaf size $= -1$, $\lambda = 0.01, \gamma = 1$
- CR: nb estimators$= 20$, max depth$= 10$, $\pi = 0.9$, $\pi_C = 0.9$

The code and the results can be found at https://github.com/anoxai/counterfactual_rules.