# OpenReview forum: "Rethinking Counterfactual Explanations as Local and Regional Counterfactual Policies"
_NeurIPS.cc/2022/Conference — NeurIPS 2022 Submitted_

### Official Review · Reviewer_b9vt · 2022-06-29

**Rating:** 6
**Confidence:** 4
**Soundness:** 3 good
**Presentation:** 3 good
**Contribution:** 3 good

**Summary:**

The authors propose to counterfactual rules instead of counterfactual samples as explanations -- i.e. the explanation is a (set of) rule(s) instead of single sample (counterfactual).

**Questions:**

None

**Strengths And Weaknesses:**

Good paper!

Only minor issues:
I think an illustration inf the form of a 2d data set + decision boundary might help even more to understand Local and Regional Counterfactual Rules.
I am bit concerned about generating rectangles for high-dimensional spaces (i.e. curse of dimensionality) -- maybe the authors could briefly comment on this.
If I understood it correctly, the users always used a LightGBM as a black-box model that is going to be explained. Since the authors propose a model agnostic method, I think evaluating on a single model only is enough -- the authors should consider several highly different models for their empirical evaluation.

---

> ### Author Response · Authors · 2022-08-02
> **Response: Reviewer b9vt**
>
> We really appreciate that you acknowledge our contributions and found our paper “Good.”. We thank you for the time you took to review the paper.
>
> > Q:  I think an illustration inf the form of a 2d data set + decision boundary might help even more to understand Local and Regional Counterfactual Rules.
>
> We will add a simple 2D dataset (e.g., moon dataset) in the Appendix, where we can observe the decision boundary and the Local and Regional Counterfactuals rules.
>
> > Q: I am bit concerned about generating rectangles for high-dimensional spaces (i.e. curse of dimensionality)
>
> We agree that generating good hyperrectangles is a very difficult problem in high dimensions. However, in our approach, we start by reducing the dimension by selecting the right directions and searching the hyperrectangles only in these directions. Moreover, we take advantage of the partition already given by the Random Forest or the Projected RF. Thus, finding the good hyperrectangles consists of finding the compatible leaves of the forest contrary to AReS or CET that search randomly for the good rectangles in all directions.
>
> > Q: If I understood it correctly, the users always used a LightGBM as a black-box model that is going to be explained. Since the authors propose a model agnostic method, I think evaluating on a single model only is enough
>
> LGBM and other tree ensembles are well approximated by Random Forest, but we have also tested on different models (typically a Neural Network, Tabnet [1]). They are well approximated, and we can obtain counterfactuals in both cases. We will give the results in the Appendix.
>
> [1] .Sercan Ö. Arik, Tomas Pfister. TabNet: Attentive Interpretable Tabular Learning.

---

### Official Review · Reviewer_VVsi · 2022-07-02

**Rating:** 3
**Confidence:** 5
**Soundness:** 1 poor
**Presentation:** 2 fair
**Contribution:** 1 poor

**Summary:**

The paper aims to address multiple problems with counterfactual
explanations that consist of single points, by replacing them by
rectangular regions that switch the class $f(x)$ of an input x with high
probability. These rectangular regions can then be communicated in the
form of rules, which are called Local Counterfactual Rules. There is
also an extension called Regional Counterfactual Rules, which explains
regions instead of single inputs x.

The definitions of the local and regional counterfactual rules rely on a
certain conditional distribution of the inputs, which therefore has to
be estimated very accurately. An important observation of the paper is
that this estimation can be made computationally efficient if $f$ is first
approximated by a random forest $\hat{f}$.

Finally there are experiments that compare the new methods to existing
methods on several data sets.



**Questions:**

The limitations of the paper seem very fundamental, and would require a
significant revision to address.


**Limitations:**

Limitations of the approach have not been sufficiently addressed, as
discussed above. In particular, it is not clear when it can be expected
to work and when not, or how to detect whether it can be applied for a
particular type of data.



**Strengths And Weaknesses:**

The structure of the paper is to discuss a series of limitations of
existing methods in the literature, and then to posit new definitions
for Local and Regional Counterfactual Rules. Unfortunately, however,
there is very little motivation for why the new definitions should be
exactly the way they are, and what their strengths and weaknesses are.
And they do strike me as having at least the following significant
weaknesses (I will focus on Local Counterfactual Rules here):

* Definition 3.3 of Local Counterfactual Rules only provides a necessary
  requirement that the rectangles should satisfy, so it is in fact not
  a full definition.
* The rectangles produced by Local Counterfactual Rules may
  actually be *implausible*, i.e. definition 3.3. leaves open the
  possibility that P(C_S(x;y*)) is arbitrarily small, because it is only
  a requirement about a probability conditional on C_S(x;y*).
* The approach critically relies on estimating a conditional probability
  for x. There should be a discussion on when this is reliable. For
  instance, this cannot work well when explaining inputs x from
  low-probability regions for which the exact distribution cannot be
  estimated very well, or if the true distribution is very irregular, or
  if the input dimension is large. The conditional probability is also
  estimated for each possible rectangle C_S(x), which will inflate the
  estimation error.
* It is not clear how a user is supposed to obtain recourse from hearing
  a rectangular region. Definition 3.3 implies that, if the user picks a
  point from the region according to the true distribution of x, then
  they will obtain recourse with high probability. But a) the user does
  not know the true distribution of x, and b) even if they did, it would
  seem unreasonable to expect them to modify their features in
  accordance with this distribution. The introduction (line 98) claims
  that their approach "partly answer[s] the problem of noisy responses
  to 99 prescribed recourses" identified by Pawelczyk et al., 2022, but
  this claim is not backed up.

Presentation:

The presentation is sufficient, but clarity could still be improved
and claims could be substantiated better. For example:
- There are multiple cases where terms are unclear or become clear only
  later, especially in the introduction. For example, line 51 refers to
  "the best promising directions", but it is not clear by which measure
  'best' is determined.
- In line 34 it is claimed that solutions from prior work are "not
  entirely satisfactory", but it is not specified in which way they are
  not satisfactory.
- Definition 3.2 confuses the distinction between a "Divergent
  Explanation" and a "Minimal Divergent Explanation".

---

> ### Author Response · Authors · 2022-08-02
> **Response 1/3: Reviewer VVsi**
>
> We are very grateful for the time taken to elaborate this detailed review, and we think that part of the critics/limitations raised may have been generated by a possible misunderstanding of our work.
>
> First of all, in the given summary, it seems that we are only trying to solve the plausibility problem (see below).
>
> > Summary: "The paper aims to address the problem that counterfactual explanations can be implausible, by which it means that they are too different from typical samples from the distribution of feature vectors x. To do this it proposes to replace counterfactuals, which are single points, by whole rectangular regions that switch the class of the input with high probability. These rectangular regions can then be communicated in the form of rules, which are called Local Counterfactual Rules. There is also an extension called Regional Counterfactual Rules, which explains regions instead of single inputs."
>
> However, we have highlighted in Section 1 and 2 that we propose rules to synthesize the diverse Counterfactual Explanations given by the classic methods, find stable regions (not close to decision boundaries) to ensure robustness to perturbations. In addition, these rules allow us to have a global picture of the model to detect certain patterns (e.g. application in fairness) while being as interpretable as possible by guaranteeing sparsity. Our methods rely on a statistical estimator (with asymptotic guarantees) and not on heuristics or constrained optimization like classical methods. This also answers the question raised about the little motivation of our methodology.
>
> > Q: It is stated that "The definitions of the local and regional counterfactual rules rely on the distribution of the inputs which therefore has to be estimated very accurately. This is done using several variants of random forests." An important weakness is the fact that the "the approach critically relies on  estimating the true distribution of x via random forests."
>
> This is not what we do. Our objective is to find counterfactual rules for $(\boldsymbol{X}, Y)$ or $(\boldsymbol{X}, f(\boldsymbol{X}))$ by using a Random Forest proxy $\hat{f}$ learnt from the data set $\{(x_i,y_i)\}$ or $\{(x_i,f(x_i))\}$. We need the random forest proxy $\hat{f}$ because our algorithm uses the tree structure for computing the rules efficiently. This means that we do not estimate the law of the feature vector $\boldsymbol{X}$, but only the conditional law $Y\vert \boldsymbol{X}$ (or $f(\boldsymbol{X})\vert \boldsymbol{X}$) with a standard Random Forest. Among others, we need to compute the Counterfactual Decision Probability  $P(f(\boldsymbol{X}) \in \mathcal{Y}^\star \left| X_{\bar{S}}=x_{\bar{S}}\right)$. While $P(f(\boldsymbol{X}) \in \mathcal{Y}^\star)$ can be easily computed for any ML model with Monte-Carlo for instance, the computational challenge is for the conditional probability. A direct way would be to compute the conditional probability $P(X \vert X_{\bar{S}}=x_{\bar{S}})$ and to integrate $f(\boldsymbol{X})$ with respect to this conditional probability ; or to be able to simulate from this conditional probability, and use a Monte Carlo approximation. We agree that such an approach have to face with a very challenging inferential problem of learning the distribution of $\boldsymbol{X}$ and all the conditional distributions $X_S \vert X_{\bar{S}}$, for all the coalition of variables $S$. Another approach would be to estimate another model $\hat{f}(X_{\bar{S}})$, but this would be very costly. As a consequence, we do not use these approaches, and we focus on approximating Random Forests. Indeed, our approach heavily relies on the use of Random Forests and the so-called "Projected Random Forest trick", introduced in [1] that permits to extract an efficient estimator of $f(\boldsymbol{X}_{\bar{S}})$ directly from the Random Forest $\hat{f}(\boldsymbol{X})$.
>
> [1]. Clément Bénard, Gérard Biau, Sébastien Da Veiga, and Erwan Scornet. Shaff: Fast and consistent
> 409 shapley effect estimates via random forests. arXiv preprint arXiv:2105.11724, 2021a.

---

> > ### Author Response · Authors · 2022-08-02
> > **Response 2/3: Reviewer VVsi**
> >
> > > Q: There should be a discussion on when this is reliable. For instance, this cannot work well when explaining inputs x from low-probability regions for which the exact distribution cannot be estimated very well, or if the true distribution is very irregular, or if the input dimension is large. It should also be discussed how sensitive the definition is to estimation errors in the distribution of x.
> >
> > We have implicitly given the assumptions on the law of X under which our approach works. It should be noted that all our methods are based on the Random Forest, the estimators derived from it, and in particular, the Conditional Decision Probability (CDP). We have shown in Section 4 that estimating the CDP is equivalent to estimating the (Same Decision Probability) SDP. In section 4 of [1], we can find all the assumptions on the distribution of X for approximating these quantities using the Random Forest. Although these are only asymptotic guarantees, to the best of our knowledge these are the best results that can be found for estimators based on Random Forests. However, we will put the assumptions more explicitly in the latest version.
> >
> > > Q: Definition 3.3 of Local Counterfactual Rules only provides a necessary requirement that the rectangles should satisfy, so it is in fact not a full definition
> >
> > We apologize because this comment may have been induced by an oversight: we forgot to mention in the definition that we consider only Rules $C_S(\boldsymbol{x};\mathcal{Y}^\star)$ satisfying (3.1) *with maximal volume* hyperrectangle. It is indicated in line 176, but this should be part of the definition. It is how the Local-CR are computed in the paper. We will change the definition accordingly. Nevertheless, we consider changing maximal volume (straightforwardly computed with the Lebesgue measure) to maximal probability $P_{\boldsymbol{X}} (C_S(\boldsymbol{x};\mathcal{Y}^\star))$.
> > Consequently, this change should also answer the following comment about the size of that $P(C_S(x;y*))$ that can be arbitrarily small. In addition, we emphasize that no mathematical definition of a Counterfactual Rule has been proposed before (AReS or CET), it is the first attempt, and all methods rely on comparisons based on accuracy, coverage, and sparsity to select the best rule.
> >
> > > Q: "It is not clear how a user is supposed to obtain recourse from hearing a rectangular region"
> >
> > One objective of our approach is to give a flexible way to change the decision: a user can pick any feasible point for him in the region $C_S(\boldsymbol{x};\mathcal{Y}^\star)$, the rule guarantees a high probability of changing the decision (greater than $\pi_C$). In some applications, the user can have domain-specific expertise that permits him to choose a feasible point. Therefore, it does not need to know the distribution, he just needs $C_S(\boldsymbol{x};\mathcal{Y}^\star)$, that has been computed beforehand. For the sake of completeness and comparisons with the closest alternative approaches (AReS and CET), we propose a simulated annealing approach for sampling a single recourse in $C_S(\boldsymbol{x};\mathcal{Y}^\star)$. Our methodology is described in the Experiments Section (but we propose to introduce and discuss our algorithm in Section 5 (as a Section 5.2) in the final version) for generating this single recourse strategy derive from the rule. With this methodology, we do not require to know the distribution of $X$, but we use only an isolation forest to avoid generating outliers.
> >
> >  > Q: The introduction (line 98) claims that their approach "partly answer[s] the problem of noisy responses to 99 prescribed recourses" identified by Pawelczyk et al., 2022, but this claim is not backed up.
> >
> > As we stated in Lines 96-99, by construction, our methods are stable because they give a range of values that guarantee a high probability of changing the decision as long as the perturbations are within our ranges. Thus, since in [2], they just add small Gaussian noises, the perturbed vectors most likely stay in the intervals. We also conducted the experiment of [2]: We normalize all the datasets so that $x \in [0, 1]$, we add gaussian noise ε to the prescribed recourse, where $ε \sim N (0, \sigma^2)$ and $\sigma^2 \in \{0, 0.01, 0.025, 0.05\}$. We compute the *Accuracy: the fraction of unseen instances where the action and perturbed action lead to the same output*. We computed the *Accuracy* for COMPAS, Diabetes, and used the simulated annealing approach with the Local-CR of Section 6 to generate the actions. The *Accuracy* for the different perturbations $\{0.01, 0.025, 0.05\}$ are: Compas$=[0.989,0.988, 0.988]$, and Diabetes$=[0.966,0.97, 0.969]$ respectively.
> >
> > [1]: Salim I Amoukou and Nicolas JB Brunel. Consistent sufficient explanations and minimal local rules for explaining regression and classification models. arXiv preprint arXiv:2111.04658, 2021
> >
> > [2] .Martin Pawelczyk et al. Lakkaraju. Algorithmic recourse in the face of noisy human responses, 2022.

---

> > > ### Author Response · Authors · 2022-08-02
> > > **Response 3/3: Reviewer VVsi**
> > >
> > > > Q:  "it is not clear when it can be expected to work and when not, or how to detect whether it can be applied for a particular type of data."
> > >
> > > As said before, the theoretical assumptions of our approach are the same as the ones used in [1] to estimate the Same Decision Probability (SDP). Note that, as in prior works (AReS, CET), our approach cannot guarantee to change the output for all individuals. However, contrary to prior methods, we give additional information by computing the Counterfactual Divergent Probability: in practice, when this probability is high, the rule always changes the decision. The probability of changing the decision is controlled by the hyperparameter $\pi_C$ as well as the approximation errors of the estimators.
> > >
> > > On the other hand, we mentioned in Lines 400-401 that our methods, being based on tree-based models, work mainly for tabular data.
> > >
> > > Finally, we thank the reviewer for the suggestions to improve the paper's clarity. We will take this into account in the final version.
> > >
> > > [1]: Salim I Amoukou and Nicolas JB Brunel. Consistent sufficient explanations and minimal local rules for explaining regression and classification models. arXiv preprint arXiv:2111.04658, 2021

---

> > > ### Comment · Reviewer_VVsi · 2022-08-09
> > > **Re: Response 2/3: Reviewer VVsi**
> > >
> > > "We have implicitly given the assumptions on the law of X under which
> > > our approach works."
> > >
> > > I cannot find these in the paper. The reference to [1] is very helpful,
> > > but notice that asymptotic consistency of an estimator does not mean
> > > that it is always good in practice. By analogy: otherwise all
> > > classification tasks could be solved by K-nearest neighbour. There
> > > should therefore be some (empirical or theoretical) evaluation of the
> > > proposed estimator in the paper, as well as some discussion on the
> > > sensitivity of the final explanations to estimation errors in the
> > > estimator.
> > >
> > >
> > > "A user can pick any feasible point for him in the region $C_S(x)$"
> > >
> > > The definition of CDP does not guarantee anything about individual
> > > points in a rectangle, only about points selected at random according to
> > > the conditional distribution, which may be highly non-uniform. So the
> > > proposed approach of picking any feasible point does not give the user
> > > any guarantees.
> > >
> > > "As we stated in Lines 96-99, by construction, our methods are stable
> > > because they give a range of values that guarantee a high probability of
> > > changing the decision as long as the perturbations are within our
> > > ranges."
> > >
> > > As explained in the previous point, this is not what your methods
> > > guarantee. But your experiment would indeed be sufficient to back up the
> > > claim if added to the paper.

---

> > > > ### Author Response · Authors · 2022-08-09
> > > > **complementary response**
> > > >
> > > > ** "notice that asymptotic consistency of an estimator does not mean that it is always good in practice": we do agree with the reviewer that the consistency is not the definitive answer, and that the rate of convergence is a much better answer. Nevertheless, we think that we already provide several new concepts (Counterfactual Decision Probability, A computational efficient + consistent estimator, Local Counterfactual Rules and Local Regional Rules) and numerous comparisons with the (best) "State-of-Art" related approach. For space constraints and for having a focused paper, we will address the limitations due the quality of estimations and their impacts in a subsequent work, and indicate them in conclusion.
> > > >
> > > > ** As said before, we do not require to select the counterfactual in the rectangle by following the distribution of X. The rectangle already contains the information (in a way they are a summary of the conditional distribution) and the user does not need to have additional knowledge of the distribution of the data. As it is remarked by reviewer VVsi, we do not provide guarantee for any point in the rectangle, and this is why when we need to select only one point in a rectangle, in propose to use an isolation forest (learnt previously) that can avoid having to different distribution from the original one. Giving a guarantee will be addressed in a subsequent work.

---

> > ### Comment · Reviewer_VVsi · 2022-08-09
> > **Re: Response 1/3: Reviewer VVsi**
> >
> > I thank the authors for their extensive response to my criticisms.
> >
> > * My summary was indeed too narrow: it is clear that the motivation for
> >   the paper is not just the plausibility problem, but also the other
> >   reasons given by the authors in their response, like robustness to
> >   perturbations.  I have updated the review.
> >
> > * About estimating the law of X vs the law of Y|X: the crucial object
> >   that is estimated in CDPs is a conditional probability in terms of X,
> >   so my criticisms all apply. The authors are correct, however, that
> >   this conditional probability is estimated with estimators derived from
> >   random forests rather than "via random forests". I have reworded my
> >   review more carefully.

---

> > > ### Author Response · Authors · 2022-08-09
> > > **Inference challenge for CDP**
> > >
> > > We thank reviewer VVsi for the discussion and the new comments. We agree that learning conditional distributions $Y\vert X$ is a simpler situation than learning all the conditional distributions $X_S \vert X_{\bar{S}}$. Indeed for classification, it is the same inferential problem of supervised learning. Concerning regression, the problem becomes a quantile regresssion problem which is also relatively well-known supervised problem.
> > > We have shown that the counterfactual problem can be addressed by using standard supervised learning tools. The problems raised by the reviewer are not specific to our method, but they are criticisms to the limitations of tree-based models. We agree that there might be some uncertainties when estimating the rules, but we thing they can be addressed with appropriate development or adaptation of Random Forests models. Typically, Random Forests tend to be well-understood  as their uncertainties , see for instance [1], [2]
> > > At the opposite, other methods that need to estimate all the conditional densities (in particular mixed data) are exposed to high theoretical or computational challenges (as sota methods like [3,4,5] for instance) does not have such kind of consistency results.
> > >
> > > [1] V-statistics and Variance Estimation. Z Zhou, L Mentch, G Hooker J. Mach. Learn. Res. 22, 287:1-287:48
> > > [2] Athey, Susan, Julie Tibshirani, and Stefan Wager. Generalized Random Forests. Annals of Statistics, 47(2), 2019.
> > > [3] Variational Autoencoder with Arbitrary Conditioning
> > > [4] ACFlow: Flow Models for Arbitrary Conditional Likelihoods
> > > [5] Arbitrary Conditional Distributions with Energy

---

### Official Review · Reviewer_CFsf · 2022-07-12

**Rating:** 6
**Confidence:** 3
**Soundness:** 3 good
**Presentation:** 2 fair
**Contribution:** 3 good

**Summary:**

To explain machine learning models, the paper proposes learning counterfactual example rules (regions) rather than generating individual counterfactual examples. This has benefits in stability and avoids the variance of producing individual examples.

**Questions:**

* Can the approach from Anchors be used in a similar way to solve the CF regions problem?
* If I understand correctly, the output policies/regions may not be 100% correct (i.e., may not flip the label). Wouldn't this violate the expectation of a counterfactual explanation?
* Can you provide some examples of the policy learnt by the model, e.g., on Compas? A qualitative analysis of the regions and their interpretability will help.

**Limitations:**

The provided policies/regions may not be 100% correct. That is a limitation.

**Strengths And Weaknesses:**

### Strengths
* Useful formulation to think of counterfactual regions versus examples.
* Technically sound method using random forests that avoids discretization parameters in past work
* Evaluation is reasonable and compares to past work.

### Weaknesses
* The writing can be improved. The idea of using Random Forests and ignoring Sbar features is simple enough. It will be nice if the paper can provide intuitive examples as in Figures 2,3,4 before presenting the technical details.
* possibly similarities to the Anchors paper that has not been discussed https://homes.cs.washington.edu/~marcotcr/aaai18.pdf

The paper solves a well-motivated problem. Counterfactual example methods, by their nature, can be unstable and lack guarantees. Providing rules or sets of feature values instead of individual examples is an important advance. I wish the authors had engaged more with explanations literature outside CFs that does propose such rules (e.g., the Anchors paper linked above).
Other than that, I think the paper is technically strong and I like the use of Random Forest as an estimation tool given its non-parametric properties. The writing can be improved to make it more intuitive. It will help to describe section 5 in an algorithm or a table to understand the different steps. It looks like there are multiple steps in the optimization and it is unclear how they interact.

---

> ### Author Response · Authors · 2022-08-02
> **Response: Reviewer CFsf**
>
> We really appreciate that you acknowledge our contributions and finding our ideas “technically strong” and "solve a well-motivated problem". We appreciate your specific suggestions, which will help us improve our paper's quality. We thank you for the time you took to review the paper.
>
> > Q: Can the approach from Anchors be used in a similar way to solve the CF regions problem?
>
> Yes. Indeed, ARes [1] used the approach from Anchors to solve the CF regions problem. Both start by discretizing/binning the variables and then sample randomly among the bins until they find a rule that satisfies some constraint e.g. coverage, accuracy, and sparsity.
>
> > Q: If I understand correctly, the output policies/regions may not be 100% correct (i.e., may not flip the label). Wouldn't this violate the expectation of a counterfactual explanation?
>
> It is also the case for other models (AReS, CET). However, with our methods, as prior information, we compute the Counterfactual Divergent Probability: in practice, when this probability is high, the rule always changes the decision. The probability of changing the decision is controlled by the hyperparameter $\pi$ as well as the approximation errors of the estimators.
>
> > Q: Can you provide some examples of the policy learnt by the model, e.g., on Compas? A qualitative analysis of the regions and their interpretability will help.
>
> For instance, If $x =$\{age=30, juv_fel_count=0, juv_misd_count=0, juv_other_count=0, priors_count=0, race:African-American = 0, race:Asian = 0, race:Caucasian = 0, race:Hispanic= 1, race:Native American = 0, race:Other = 0, c_charge_degree:F = 0, c_charge_degree:M = 1, gender=1\}, the output is 1. We can change the decision with probability CDP=0.92 by applying this rule $C_S(x)$ = \{juv_misd_count = 1 and prior count >= 4.5\}.
>
> As suggested by the Reviewer, we will add in the final version a qualitative analysis of the Counterfactual Rule learnt on Compas, a more detailed discussion about the similarities with Anchors and the other methods (ARes, CET), and a pseudo-code algorithm for the complete procedure.
>
> [1]: Salim I Amoukou and Nicolas JB Brunel. Consistent sufficient explanations and minimal local rules
> 407 for explaining regression and classification models. arXiv preprint arXiv:2111.04658, 2021

---

### Meta-Review · Area_Chair_7xJj · 2022-08-30

**Recommendation:** Reject
**Confidence:** Certain

**Metareview:**

This paper highlights a series of limitations of existing methods in the literature on algorithmic recourse (e.g., recourses are not implemented exactly and are often noisy), and posits new definitions for Local and Regional Counterfactual Rules and proposes a novel algorithmic framework to learn them. All the reviewers opine that there is very little motivation for why the new definitions should be exactly the way they are, and what their strengths and weaknesses are. In addition, it is unclear when we can expect the proposed approach to provide a good estimate of the criterion. Furthermore, the problems highlighted in this work have been explored in several recent works (e.g., Dominguez-Olmedo et. al., ICML 2022, On the adversarial robustness of causal algorithmic recourse; Pawelczyk et. al., 2022, Let Users Decide: Navigating the Trade-offs between Costs and Robustness in Algorithmic Recourse). We encourage the authors to address the aforementioned aspects, discuss related works, and also compare the proposed approach with these works.


**Award:**

No

---

### Decision · Program_Chairs · 2022-09-14

Reject